# Inchworm Robots Utilizing Friction Changes in Magnetorheological Elastomer Footpads Under Magnetic Field Influence

**DOI:** 10.3390/mi16010019

**Published:** 2024-12-26

**Authors:** Yun Xue, Chul-Hee Lee

**Affiliations:** Department of Mechanical Engineering, Inha University, Incheon 22212, Republic of Korea; xueyun@inha.edu

**Keywords:** magnetorheological elastomers, inchworm-inspired robots, friction coefficient

## Abstract

The application of smart materials in robots has attracted considerable research attention. This study developed an inchworm robot that integrates smart materials and a bionic design, using the unique properties of magnetorheological elastomers (MREs) to improve the performance of robots in complex environments, as well as their adaptability and movement efficiency. This research stems from solving the problem of the insufficient adaptability of traditional bionic robots on different surfaces. A robot that combines an MRE foot, an electromagnetic control system, and a bionic motion mechanism was designed and manufactured. The MRE foot was made from silicone rubber mixed with carbonyl iron particles at a specific ratio. Systematic experiments were conducted on three typical surfaces, PMMA, wood, and copper plates, to test the friction characteristics and motion performance of the robot. On all tested surfaces, the friction force of the MRE foot was reduced significantly after applying a magnetic field. For example, on the PMMA surface, the friction force of the front leg dropped from 2.09 N to 1.90 N, and that of the hind leg decreased from 3.34 N to 1.75 N. The robot movement speed increased by 1.79, 1.76, and 1.13 times on PMMA, wooden, and copper plate surfaces, respectively. The MRE-based intelligent foot design improved the environmental adaptability and movement efficiency of the inchworm robot significantly, providing new ideas for the application of intelligent materials in the field of bionic robots and solutions to movement challenges in complex environments.

## 1. Introduction

Nature has become a rich source of inspiration for robot design. From insects to large animals, their extraordinary adaptability and efficiency in navigating complex terrains have opened new avenues for solving locomotion and manipulation challenges in complex environments [1]. Among biologically inspired robots, the inchworm has attracted particular interest in the robotics community. The simple yet effective body structure and unique locomotion patterns of the inchworm allow it to navigate flexibly through a variety of complex environments. This ability has generated significant interest in the potential applications of inchworm-inspired robots, particularly in confined space operations and pipeline inspections [2]. The ability of the inchworm to traverse narrow passages and adhere to surfaces with different inclinations makes it an ideal model for robots designed to operate in challenging industrial environments.

Inchworm robotics research falls into two main categories: soft and rigid robots [3,4]. Each approach has unique advantages in replicating the locomotor capabilities of inchworms but poses different challenges for soft and rigid wake robots. Rigid inchworm robots have been constructed to resist bending and deformation and excel in scenarios that require heavy lifting or maneuvering in harsh environments. Their robust construction allows them to withstand enormous loads and unfavorable conditions, making them ideal for industrial applications where durability is critical [5]. Nevertheless, they are not flexible enough to adapt to contact surfaces. Soft robotic systems, however, exhibit greater flexibility and adaptability by altering contact surface properties to accommodate different surface conditions, more closely mimicking the flexibility of living organisms [6]. These robots can deform and adapt to their environment, allowing for a safer interaction with delicate objects or for human-centered applications. On the other hand, soft robots often face challenges in generating high intensity or maintaining precise positioning, which may limit their applicability in certain situations.

Therefore, hybrid soft–rigid robots offer greater benefits in terms of resistance to bending and deformation while adapting to various surface conditions by varying the contact surfaces [7]. Hybrid soft–rigid robotic systems aim to combine the best attributes of both approaches, balancing the strength and precision of rigid systems with the adaptability and safety of soft systems. Many scholars have examined the application of smart materials in hybrid soft–rigid looper robots. Zhang et al. [8] proposed a hybrid looper robot with variable stiffness characteristics, which was controlled by stiffness-enhanced hybrid actuators to achieve an enhanced load-bearing capacity while maintaining the flexibility of soft robots, demonstrating great utility in field exploration and cargo transportation. On the actuation side, Sean Thomas et al. [9] developed an innovative hybrid inchworm robot. The robot used an SMA-driven mechanical oscillator based on a pliable structure as an actuator, combined with a novel magnetic locking system to achieve autonomous locomotion without the need for any control strategy or electronic devices. Zhong et al. [10] explored the use of electroactive polymers (EAPs) as a soft part actuator source combined with a rigid skeleton to achieve multi-surface climbing functionality using a 3D scissor mechanism and electrostatic adhesion to reduce robot deployment requirements. Magnetorheological elastomers (MREs) are smart materials composed mainly of silicone rubber and magnetic fillers [11]. The stiffness and modulus of this rubbery polymer containing magnetic particles can be varied in a magnetic field [12], which regulates the friction coefficient of the MRE surface [13], enabling the inchworm robots to perform crawling and climbing. MREs have been used in robotics mainly as flexible exoskeletons [14], joint connectivity [15], and vibration-damping applications [16], facilitating the flexible adjustments and rapid deformation of the robots. These properties make MREs highly applicable in inchworm robots. MREs have attracted considerable interest for their promising applications in dynamic systems, owing to their controllable mechanical properties [17,18]. In the application of MREs in climbing robots, Qi and Zhao [19,20] used MRE as a shape-programmable material to fabricate an inchworm bionic structure with walking, swimming, and grasping functions. Hong et al. [21] used the high magnetic permeability of MREs compared to non-magnetic materials to make it work as a footpad of a magnetic foot, which enhances the retention force in the shear direction. In addition, the friction properties of MREs are still a great challenge for the application in the field of inchworm robotics.

Considering this background and the challenges that exist in this field, this research aimed to develop a novel, intelligent inchworm robot that combines traditional mechanical design principles with advanced smart materials. The main goal of this work was to utilize the frictional properties of magnetorheological elastomers to enhance the environmental adaptability and locomotor efficiency of an inchworm robot, which is particularly beneficial for achieving crawling locomotion.

## 2. Design of Inchworm Robot

The robot uses magnetorheological elastomers as footpads with 3D-printed front and rear leg structures and combines stepper motors, gears, and linkage systems as the driving core. Electromagnets control the surface friction of the magnetorheological elastomer crawling footpads. The friction-change behavior of MREs under a magnetic field has been widely studied. Lian et al. [22] found through rolling friction experiments that the addition of a magnetic field can increase the rolling friction coefficient at different rolling slip rates, which is used in anti-locking systems. Li and Nguyen examined the friction model of MREs, and Li analyzed the friction model of MREs with magnetic field adjustments [23,24]. Li reported that under magnetic field conditioning, the variation of the friction coefficient was as high as 41%, and Nguyen’s model was favorable for its design of MRE-based vibration devices. Nevertheless, the application of this MRE phenomenon in an inchworm robot is still largely unexplored [25]. This paper reports on a new approach to motion control through magnetic field modulation by integrating MRE-based footpads into a bionic inchworm robot. This study comprehensively evaluated the magnetic-field response properties of magnetorheological elastomers in different environments by systematically testing the friction of the front and back legs on different surfaces. The mechanism of friction change of the material under the action of an applied magnetic field was revealed. In addition, through a comparative analysis of the crawling distance of the inchworm robot on different surfaces at the same time, the effect of the friction change of the magnetorheological elastomer on the macro-motor performance of the robot, particularly the crawling speed, was further explored in depth, and the feasibility and validity of the smart material to enhance the motion efficiency of the bionic robot in complex environments were verified. Details about the preparation of MRE materials, structural design, and motion mechanism of the robot are as follows.

### 2.1. Analysis of Materials and Surface Microscopic Properties

#### 2.1.1. Preparation of Magnetorheological Elastomers

The MREs used in this study were a silicone rubber matrix mixed with its supporting curing agent in a 50:1 ratio (SYLGARD 184, DOWSIL, Inc., Midland, MI, USA), and iron–silicon–chromium magnets with an average diameter of 15 μm were added. The filler (FSC-82-15, Changsung, Incheon, Republic of Korea) was prepared.

In the preparation process of the MRE samples, the silicone rubber, the curing agent, and carbonyl iron powder matching the silicone rubber were first weighed accurately, according to the set ingredient ratio. Three raw materials were weighed and premixed for 10 min under manual stirring to ensure a uniform dispersion of the carbonyl iron particles in the silicone rubber matrix. Subsequently, the mixture was placed in a mechanical stirring device and stirred continuously at 280 rpm for 30 min to enhance the mixing uniformity of the material, as shown in Figure 1.

After the mixing was completed, the mixture was poured into a prepared inchworm robot footpad mold and placed in a vacuum box with a vacuum of 0.2 bar. The negative pressure state was maintained for 30 min to remove the MRE mixture introduced during the stirring of any bubbles. This prevents bubbles from affecting the modulus and surface of the magnetorheological elastomer.

Finally, the degassed mixture was placed in an oven and baked at a constant temperature of 70 °C for two hours and 30 min to solidify the silicone rubber matrix. An MRE sample with a magnetic filler volume fraction of 50% concentration was prepared.

#### 2.1.2. Surface Observation Experiment of MRE Materials

The surface microstructure of the magnetorheological elastomer was observed using a digital microscope (DVM6, Leica Microsystems, Wetzlar, Germany), as shown in Figure 2. Micrographs were taken at magnifications of (i) 50×, (ii) 200×, (iii) 500×, and (iv) 1000×, and Figure 2i shows the homogeneous distribution of the surface of magnetorheological elastomers, and the same can be seen in Figure 2ii. Magnetic particle agglomerates are found to be dispersed in the elastomer substrate, and the particle density is clearly visible. Figure 2iii shows the spherical shape of the individual magnetic particles and their random embedding in the silicone substrate. Figure 2iv clearly demonstrates the interfacial bonding between the magnetic filler and the elastomeric base. A multi-scale microscopic observation demonstrated the structure of the magnetic filler and the base within the magnetorheological elastomer.

The addition of a magnetic field affects the surface properties of magnetorheological elastomers, as shown in Figure 3, where micrographs were taken (a) without a magnetic field and (b) under the influence of a 60 mT magnetic field. A computer was connected to the digital microscope to store and process the image data.

Figure 3 clearly shows a significant difference in the surface topography of the MRE sample without and with a 60 mT magnetic field. The color-coded height maps visually reflect changes in surface morphology. In the absence of an applied magnetic field, the MRE surface exhibits a clear heterogeneous topography with multiple prominent peaks (red areas) and valleys (blue areas), indicating a rough surface with large local fluctuations in height. According to the z-axis profile data, the height difference in local areas exceeded 100 μm, and the highest peak reached 216 μm.

On the other hand, the sample surface changed significantly after applying a 60 mT magnetic field. The surface became more uniform, dominated mainly by blue and green areas, and the surface roughness was reduced significantly. The z-axis profile also confirms this change. The maximum peak value drops to 196 μm. Compared with the condition without a magnetic field, the height change of the surface was reduced significantly. This surface-smoothing phenomenon can be attributed to the rearrangement of magnetic particles in the elastomer matrix under the influence of an applied magnetic field. Because ferromagnetic particles tend to form chain structures along the direction of the magnetic field lines, the surface structure is more orderly and uniform, reducing the roughness of the surface. This phenomenon shows that the magnetic field has a significant control effect on the surface morphology of MREs and can effectively improve the performance of this material in complex environments.

### 2.2. Structural Design

#### 2.2.1. Structural Design of the Inchworm Robot

The inchworm-inspired robot consists of three main components: a body, MRE footpads, and an electromagnetic control system. The body includes front legs, hind legs, electromagnets, stepper motors, gears, turntables, and transmission rods. The MRE footpad is the core component of the invention. The surface friction can be adjusted by altering the strength of the external magnetic field. The electromagnet was installed just above the footpad, and the opening and closing of the electromagnet are precisely controlled through the electromagnetic control system to change the magnetic field. This design enables the robot to adjust friction in real time, improving its adaptability and crawling efficiency in various environments.

The robot body was designed using SolidWorks and was 3D-printed using PLA material. The body is 180 mm long when contracted, 205 mm when extended, 116 mm wide, and 79 mm high. The body consists of two parts connected by a stepper motor (17HS4023, Usongshine, Shenzhen, China). The front and rear legs of the robot move in a coordinated manner through the stepper motor, gear, and transmission rod system, simulating the crawling method of an inchworm. At the same time, the front and rear foot alternating control mechanism was reasonably designed to achieve a propulsion method that met the requirements of bionic peristalsis. As shown in Figure 4, the magnetorheological elastomer pad of the inchworm robot is controlled by an electromagnet, so that it can be well adapted to wooden boards, copper plates, and PMMA.

The locomotion capability is achieved through a coordinated sequence of actuations, as shown in Figure 5. In Figure 5a, the crawling motion of the inchworm robot consists of two phases: anchor pull (i) and anchor push (ii). In the anchor-pull phase (i), the electromagnets of the front legs are turned off, while the electromagnets of the rear legs are turned on. The friction of the magnetorheological elastomer pads of the rear legs is reduced, and the front legs maintain high friction for anchoring, thereby generating a friction difference between the front and rear legs. Then, the crank-connecting rod mechanism starts to act, using the front legs as anchors to pull the rear legs forward.

In the anchor-push phase (ii), the state of the electromagnets is reversed, the front-leg electromagnets are turned on, and the rear-leg electromagnets are turned off, making the friction of the front-leg MRE pads less than the friction of the rear-leg MR elastomer pads. The rear legs are anchors, and then, the crank-connecting rod mechanism pushes the front legs forward, and they return to the posture of the anchor-pull phase. Then, the above actions are repeated, and this sequence of variable friction and mechanical drive allows the inchworm robot to crawl forward continuously.

As shown in Figure 5b, the central control unit acts as the main controller, receiving power from the 9 V power supply and controlling the mechanical drive and electromagnetic components of the inchworm robot. The mechanical drive is achieved through a stepper motor (powered by 24 V), which accurately controls the movement of the front and hind legs through a crank-connecting rod mechanism. At the same time, the central control unit controls the electromagnets of the front and rear legs. Electromagnetic control is achieved through two parallel circuits, each channel containing a 24 V power supply and a time relay system. The electromagnets control the friction state of their respective MRE footpads through magnetic field modulation, and the friction force of the magnetorheological elastomer pad is affected by a change of the electromagnet magnetic field.

#### 2.2.2. MRE Foot Integration

Two precisely designed grooves were preset on the bottom of the front and rear legs to secure four MRE footpads to the bottom of each body section. Optimizing the position of the feet ensured that the MRE footpads could achieve the maximum contact area with the surface during the movement of the robot, improving the accuracy of friction control and the stability and efficiency of the movement of the robot in complex environments. This structural design enhanced the performance advantages of MRE materials and significantly improved the ability of the robot to adapt to different surfaces, as shown in Figure 6a.

Figure 6b,c show the mechanism of friction change in MRE footpads under a magnetic field, illustrating how the application of a magnetic field affects the alignment of magnetic particles inside MRE footpads, thereby reducing friction. By applying a magnetic field, the magnetic particles are aligned inside the MRE footbed in the direction of the magnetic field, and the magnetic field magnetizes these particles to create a magnetic force (Fp) between them and attracts them to each other, leading to a reduction in the contact area and hence a reduction in friction.

### 2.3. Experimental Setup

#### 2.3.1. Setup for Friction Measurement

The friction force of the magnetorheological elastomer footpad under an applied magnetic field and no magnetic field was measured in detail using a friction tester. The results were used to illustrate its friction performance. Figure 7 presents the overall experimental diagram, showing all the test equipment for friction measurements, its practical applications, and its role in the schematic diagram. In this experiment, all measurements were taken under a fixed magnetic field strength (60 mT). The sliding stroke in the experiment was set to a fixed value (35 mm), and the sliding speed was set to 6.7 mm/s. The experiment used a sensor with a sensitivity of 0.37 mV/V (DBCM, BINGSHIN, Osan, Republic of Korea), as shown in Table 1. During the experiment, the sensor was used to monitor the change in friction between the MRE and different test surfaces in real time and recorded it through the data acquisition system. The accuracy of the experiment and the reliability of the data were ensured by taking the average value after multiple tests during the friction test. Each group of experiments was repeated three times to ensure the stability of the data and the reliability of the statistics.

#### 2.3.2. Setup for Motion Performance Evaluation

The effects of magnetic field control on the crawling distance of the inchworm robot on different surfaces, including PMMA, wooden, and copper surfaces, were evaluated. Figure 8 shows the experimental setup. The time it took for the robot to crawl this distance with or without magnetic field control was recorded using an ultrasonic sensor to ensure the accuracy and consistency of the data. Three repeated measurements were performed under each experimental condition, and the average value was taken as the final result. The specific impact of magnetic field control on improving the robot’s motion performance was quantified by comparing the crawling speeds under different surface materials and magnetic field conditions.

## 3. Results and Discussion

### 3.1. Results of Friction Measurement

The application of a magnetic field on the PMMA surface had a significant impact on the friction characteristics of the MRE foot, as shown in Figure 9a,b. The maximum friction force of the front legs decreased by approximately 9.1%, from 2.09 N without a magnetic field to 1.90 N with a magnetic field. The change in the hind legs was more apparent, and the maximum friction force decreased by 47.60%, from 3.34 N to 1.75 N. This controlled reduction in friction confirms the effectiveness of the MRE in regulating the interaction of the robot foot with smooth surfaces.

Figure 9c,d present the friction characteristics of the MRE foot on a copper plate surface. The copper plate surface exhibited higher overall friction than the PMMA surface, which may be due to the microstructure and chemical properties of the copper plate surface. After applying the magnetic field, the maximum friction force of the front legs decreased from 2.79 N to 1.95 N, and the rear legs decreased from 3.28 N to 2.64 N. The relative reduction ratios were significant, 30.1% and 19.5%, respectively. Hence, the MRE foot can maintain good magnetic responsiveness on metal surfaces, providing the possibility for these robots to be applied in industrial environments.

Figure 9e,f present the friction characteristics of the MRE foot on a wooden surface. The surface of the wood board exhibited a different friction pattern than the PMMA and copper plates. In the absence of a magnetic field, the friction on the surface of the wood board fluctuated greatly, which may be due to the natural texture and unevenness of the wooden surface. After applying the magnetic field, the maximum friction force of the front legs decreased from 2.68 N to 1.96 N, and the friction force of the hind legs decreased from 2.89 N to 2.63 N. The relative reduction ratios of the friction force were 26.9% and 9.0%, respectively. Although the absolute reduction in friction was not as significant as for the other two surfaces, the application of the magnetic field stabilized the fluctuations in friction, making the curve smoother. This feature is of great significance for the stable movement of robots in natural environments, suggesting that an MRE foot can adapt and optimize the movement performance on irregular surfaces.

Overall, Figure 9a–f show that the application of a magnetic field significantly reduced the friction in both the front and back legs, and this reduction in friction can be attributed to the arrangement of the magnetic particles within the MRE, which reduces the contact area and hence the friction. For example, in Figure 9a,b, during the anchor-pull phase, the friction of the front leg is 2.09 N without the magnetic field and 1.75 N for the back leg with the added magnetic field, which creates a friction difference between the front leg and the back leg, which makes the robot’s back leg extend forward. Whereas, in the anchor-push phase, the friction of the front leg is 1.90 N, and that of the back leg is 3.34 N, which creates a friction difference that makes the front leg of the inchworm robot extend forward. Similarly, the Figure 9c,d comparisons demonstrate the friction measurements on a copper surface, where the friction of the front leg is 2.79 N during the anchor-pull phase, which is higher than that of the rear leg, which is 2.64 N, and during the anchor-push phase, it is the rear leg, which is 3.28 N, that is higher than that of the front leg, which is 1.95 N. When on a wooden surface, as shown in Figure 9e,f, the friction of the front leg during the anchor-pull phase is 2.68 N, which is higher than that of the 2.63 N for the rear leg, and in the anchor-push phase, it is 3.89 N for the rear leg compared to the 1.96 N for the front leg. The overall reduction in friction on the different surfaces proves the adaptability and effectiveness of MRE footbeds in a variety of environments.

The MRE foot exhibited good magnetic-field responsiveness on three different surfaces and can effectively adjust the size and characteristics of friction. This adaptability enabled the MRE-based inchworm robot to move flexibly in a variety of environments, laying the foundation for future applications in complex terrains.

### 3.2. Results of Motion Performance Evaluation

Based on the changes in friction characteristics, the impact of the magnetic field on the movement speed of the robot on different surfaces was further evaluated. Applying a magnetic field significantly increased the crawling distance of the robot in a fixed time on all test surfaces, increasing the speed of movement, as shown in Figure 10. Among them, the inchworm robot achieved the longest crawling distance on the copper surface, and the crawling on the wooden surface was also comparable and improved the most.

Combining Figure 11 with Figure 10, on the PMMA surface, the crawling distance of the robot within a fixed time increased from 9.7 cm without a magnetic field to 16.526 cm with a magnetic field, and the speed increased by approximately 70.4%. On the wooden surface, the distance increased from 11.24 cm to 19.39 cm, and the speed increased by approximately 72.5%. The most significant improvement occurred on the surface of the copper plate, where the distance increased from 15.23 cm to 19.31 cm, and the speed increased by 26.8%. These results show that controlling the friction characteristics of MRE feet through magnetic fields can improve the efficiency of robot movements on various surfaces. Although the speed improvements on the wooden and PMMA surfaces were similar and larger, the absolute speed on the copper plate surface was the fastest, which may be related to its smooth surface properties and the special interaction of the MRE with the metal surface. This discovery provides an important basis for optimizing the performance of robots on different material surfaces.

### 3.3. Discussion of MRE Characteristics and Effects on Performance of Inchworm Robot

The change in friction of the inchworm robot can be attributed to the response of magnetic particles in the magnetorheological elastomer to the applied magnetic field. Silicone materials usually have excellent wear resistance. The magnetic fillers added to silicone make the magnetorheological elastomer exhibit a magnetorheological effect in the presence of a magnetic field, as well as a controllable modulus and friction. It is worth mentioning that this characteristic is reversible, that is, the magnetorheological elastomer can restore its original characteristics after the application of the magnetic field is stopped. Under the action of the magnetic field, the ferromagnetic particles in the elastomer matrix are arranged along the magnetic field lines to form an organized chain structure, which reduces the maximum peak value of the microscopic surface of the MRE. The increased contact area reduces the friction of the magnetorheological elastomer footpad under three different surface conditions in Figure 9a–f.

This change in friction and magnetorheological elastomers help robots enhance their environmental adaptability and motion efficiency. First, the magnetic particle arrangement induced by the magnetic field increases the contact interface, reduces random surface interactions, and improves motion stability. Secondly, the dynamic response of the MRE to the magnetic field can adjust the friction characteristics in real time, allowing the robot to adapt to different surface conditions during movement. Finally, the reversibility of the magnetic field effect ensures consistent performance over multiple motion cycles. And the storage modulus and hardness of the isotropic magnetorheological elastomer will increase due to the addition of the magnetic field, which will increase the mechanical properties of the magnetorheological elastomer and make it more wear-resistant, which enables it to achieve a more efficient energy transfer in the anchoring stages of the inchworm robot.

The increase in the crawling speed of the inchworm robot was due to the combined effect of the mechanical properties of the magnetorheological elastomer and the controllable friction force. The crawling of the inchworm robot relies on the difference in friction between the front and back legs. By adding a reversible magnetic field to control the friction of the magnetorheological elastomer, the friction of the front and back feet of the inchworm robot can be changed. For example, under the condition of a copper plate, where no magnetic field is added, the friction of the hind legs is constantly greater than the friction of the front legs, with a difference of 0.49, and only anchoring can be performed. After adding the magnetic field, a greater friction difference can be achieved, and the friction of the front legs and the friction of the hind legs can be controlled. The other two surfaces also show such performance. This helps the inchworm robot to move stably, and as shown in Figure 11, the crawling speed of the inchworm robot is significantly increased, because the increased friction difference enhances the motion performance of the inchworm robot.

In order to ensure that the inchworm robot can crawl stably on the surface, we selected a stepper motor that can accurately move forward to control the crawling gait of the inchworm robot. In addition, the crank-slider structure of the inchworm robot is simple in design, which reduces the free movement of the inchworm robot. The central control unit controls the electromagnet and the stepper motor at the same time, so that the electromagnets of the front and hind legs of the inchworm robot can accurately change the friction of the magnetorheological elastomer during anchor pushing and anchor pulling. This stable and controllable friction change reduces the energy loss caused by unnecessary surface interactions and reduces the possibility of slipping during crawling, thereby effectively improving the movement efficiency and environmental adaptability of the inchworm robot. Through experiments, it was found that this design enables the inchworm robot to adapt to the environment while achieving good motion performance.

## 4. Conclusions

This paper reports on the development and evaluation of a novel inchworm-inspired robot using magnetorheological elastomers (MREs) as smart footpads. This study revealed the effectiveness of MRE-based footpads in enhancing the locomotor capabilities and environmental adaptability of bionic robots. The main findings of this work are as follows.

Material properties: The fabricated MRE footpads were composed of a silicone rubber matrix with a volume fraction of 50% ferromagnetic particles, and the surface friction changed significantly under an applied magnetic field. A microscopic analysis showed that when exposed to a 60 mT magnetic field, the surface of the MREs became flatter, which explains why the friction changes after increasing the magnetic field.

Friction conditioning: Systematic friction tests on PMMA, copper, and wooden surfaces showed that friction was reduced significantly after applying a magnetic field. The most obvious effect was observed on the PMMA surface, where the friction on the front and rear legs was reduced by 9.1% and 47.60%, respectively. This controllable friction adjustment validated the potential of MREs for adaptive motion in different environments.

Movement performance: The integration of MRE footpads significantly improved the mobility of the robot on all test surfaces. After applying a magnetic field, the moving distance of the robot on PMMA, wooden, and copper surfaces within a fixed time increased by 70.4%, 72.5%, and 26.8%, respectively. Hence, MRE-based smart actuators can optimize robot motions through real-time friction control.

Environmental adaptability: consistent performance improvements across different surface materials highlight the enhanced adaptability of the robot, a critical factor in operating in complex real-world environments.

This study contributes to the field of smart material-based soft robotics by showing a new method of controlling motions through magnetically tunable friction. The successful integration of MREs in biomimetic robot design opens new avenues for developing highly adaptive robotic systems capable of navigating challenging terrains.

Future work should focus on optimizing MRE compositions to achieve a larger friction-tuning range, exploring the potential of gradient magnetic fields for finer control and investigating the long-term durability of MRE actuators under repeated cycling. Furthermore, the integration of a closed-loop control system using real-time friction feedback can enhance the autonomy and adaptability of the robot in dynamic environments.

In summary, this research lays a good foundation for developing next-generation adaptive robotic systems, bridging the gap between traditional rigid robots and fully soft robotic platforms. The capabilities of MRE-based actuators in improving motion efficiency and environmental adaptability are significant for applications in industrial inspections, search and rescue operations, and exploration missions in unstructured environments.

## Figures and Tables

**Figure 1 micromachines-16-00019-f001:**
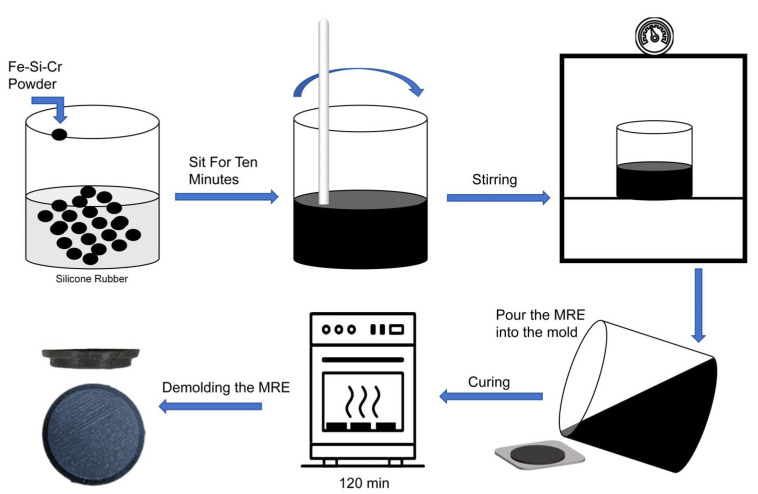
Schematic diagram of the MRE fabrication process: from mixing to molding and curing.

**Figure 2 micromachines-16-00019-f002:**
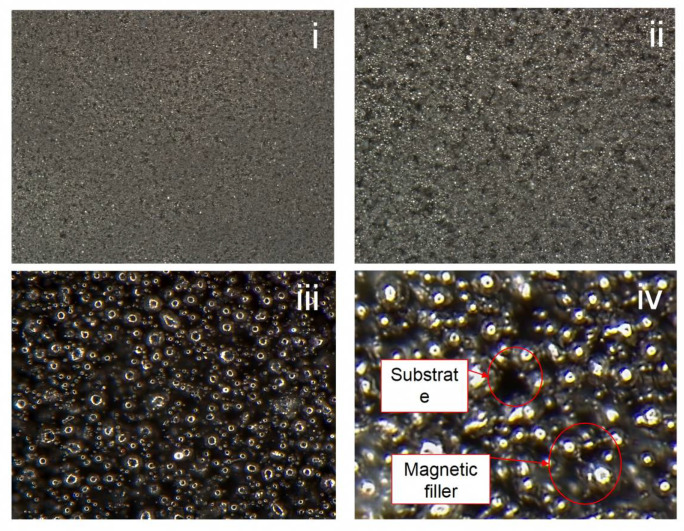
Multi-scale microscopic observations of MRE structures (**i**) at 50× magnification, (**ii**) at 200× magnification, (**iii**) at 500× magnification, and (**iv**) at 1000× magnification.

**Figure 3 micromachines-16-00019-f003:**
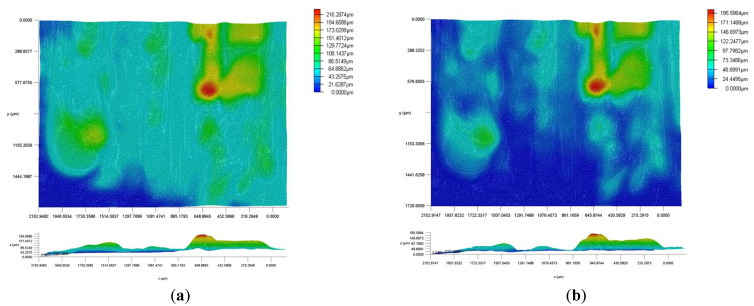
Surface topography analysis of MRE under varying magnetic field conditions. (**a**) Surface topography mapping of MRE at initial magnetic field condition. (**b**) Surface topography mapping of MRE under enhanced magnetic field intensity.

**Figure 4 micromachines-16-00019-f004:**
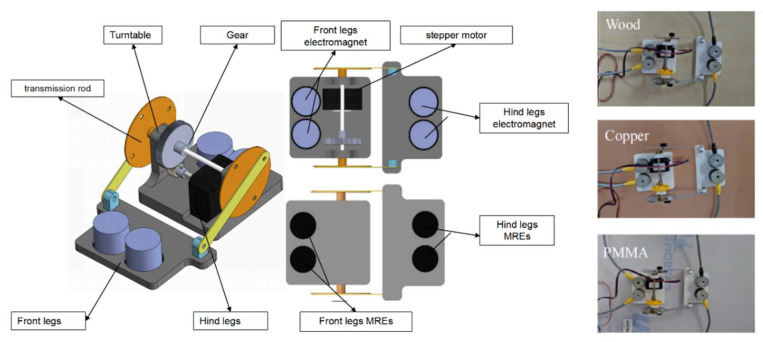
Design and components of the inchworm-inspired robot with MRE footpads.

**Figure 5 micromachines-16-00019-f005:**
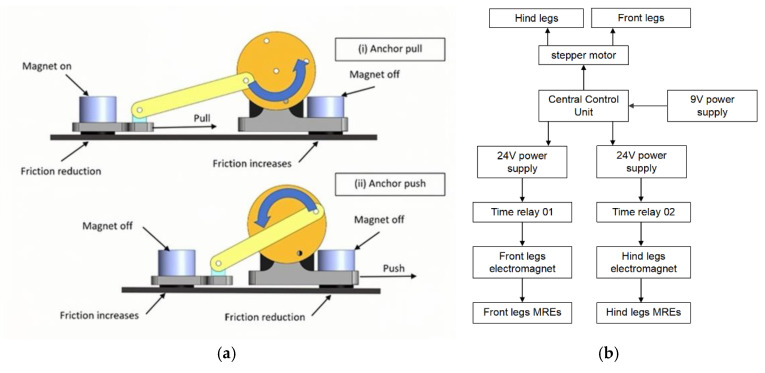
Locomotion mechanism and control system of the MRE-based inchworm robot: (**a**) Illustration of the crawling sequence showing (i) anchor pull and (ii) anchor push motions. (**b**) Hierarchical control system schematic.

**Figure 6 micromachines-16-00019-f006:**
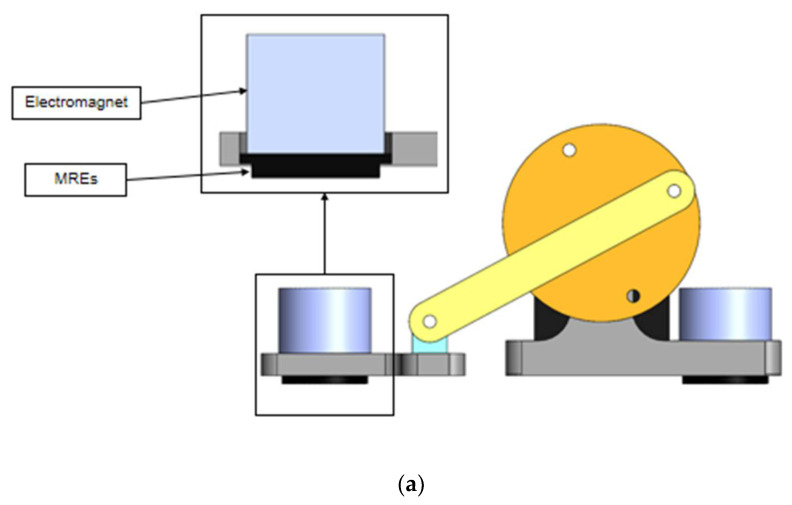
Detailed assembly of the inchworm robot foot: (**a**) Integration of the electromagnet and the MRE footpad. (**b**) Arrangement of magnetic fillers in the magnetorheological elastomer without a magnetic field. (**c**) Arrangement of magnetic fillers in the magnetorheological elastomer with a magnetic field.

**Figure 7 micromachines-16-00019-f007:**
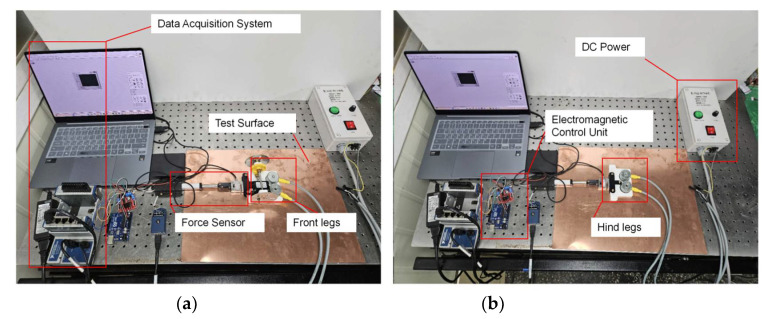
Experimental setup for friction measurements of the MRE footpads. (**a**) Front leg measurement configuration and (**b**) rear leg testing arrangement.

**Figure 8 micromachines-16-00019-f008:**
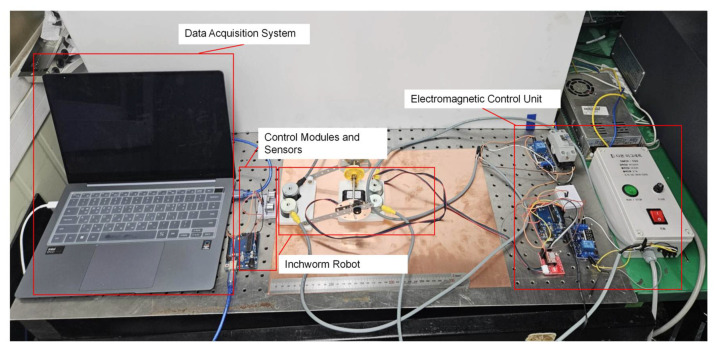
Velocity measurement setup for the MRE-based inchworm robot: integrated system with real-time data acquisition and control.

**Figure 9 micromachines-16-00019-f009:**
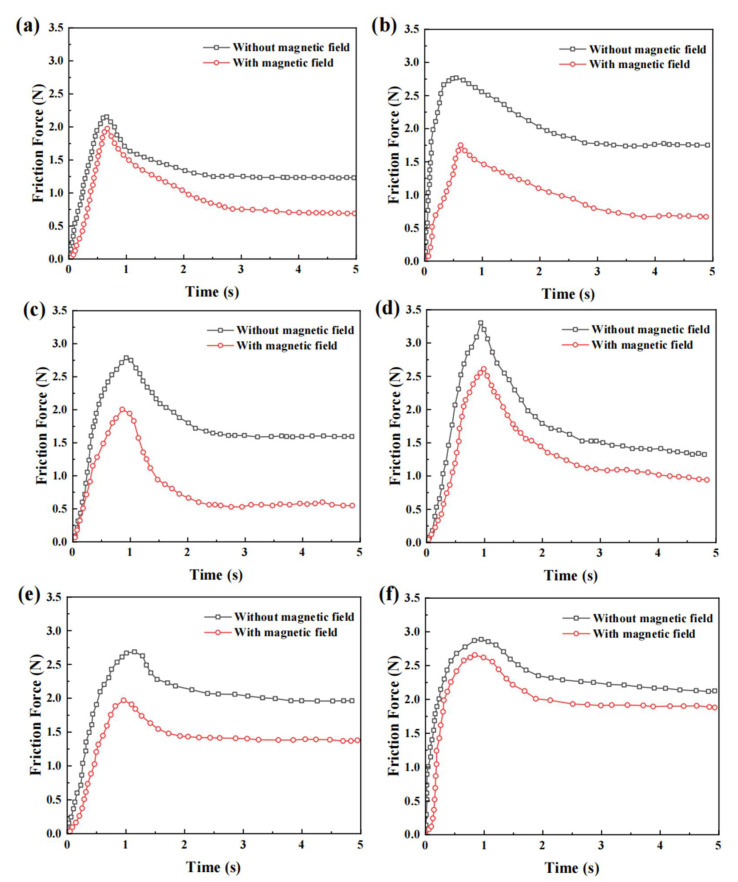
Comparative analysis of friction force in MRE footpads with and without a magnetic field. (**a**) Front leg on PMMA surface; (**b**) hind leg on PMMA surface; (**c**) front leg on copper plate surface; (**d**) hind leg on copper plate surface; (**e**) front leg on wooden surface; (**f**) hind leg on wooden surface.

**Figure 10 micromachines-16-00019-f010:**
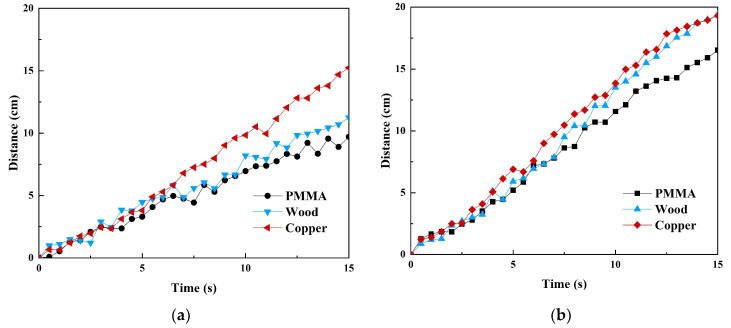
Locomotion performance of an MRE-based inchworm robot on various surfaces: (**a**) baseline translation dynamics without field activation and (**b**) enhanced locomotion performance under an applied field stimulus.

**Figure 11 micromachines-16-00019-f011:**
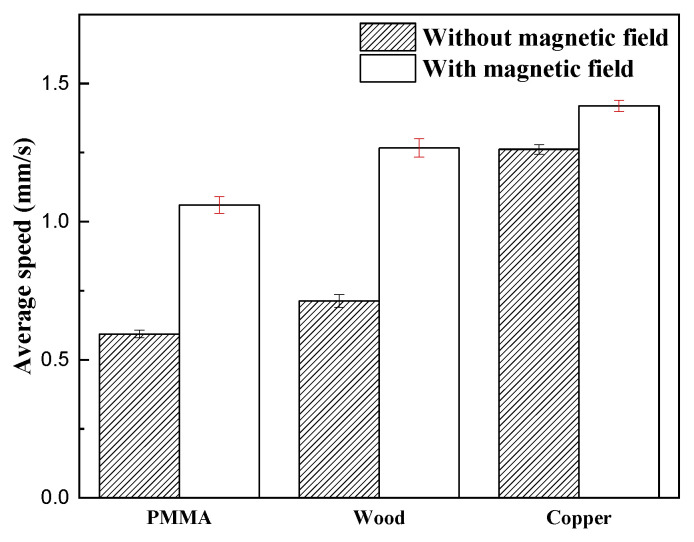
Comparative analysis of MRE-based inchworm robot average speeds on various surfaces.

**Table 1 micromachines-16-00019-t001:** Experimental parameters and operating conditions for inchworm robot.

Content	Parameter
Surface material	PMMA, copper, and wood
Test part	Front and hind legs
Speed	6.7 mm/s
Temperature	26 °C
Magnetic field strength	60 mT

## Data Availability

The original contributions presented in the study are included in the article, further inquiries can be directed to the corresponding authors.

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
