# Peer review of "Inchworm Robots Utilizing Friction Changes in Magnetorheological Elastomer Footpads Under Magnetic Field Influence"

_micromachines, 2024, doi:10.3390/mi16010019_

Round 1

Reviewer 1 Report

Comments and Suggestions for Authors

The authors utilized the frictional properties of magnetorheological elastomers (MRE) to enhance the adaptability and movement efficiency of inchworm robot. The moving distance of the inchworm robot integrated with MRE foot pads on PMMA, wood and copper surfaces within a fixed time increased by 70.4%72.5% and 26.8%, respectively. The successful integration of MRE in biomimetic robot design opens new avenues for developing highly adaptive robotic systems.

I recommend the publication of this manuscript, if the authors clearly address the following issues:

1. For a, b and c in Figure 9, the authors just simply described the measurement results of the friction force in MRE footpads with and without a magnetic field. The authors should summary and analyze the measurement results clearly for readers' understanding.

2. Authors should add a schematic diagram to explain the mechanism for the increasing of friction force under the magnetic field in part of 2.2.2. MREs Foot Integration.

3.  Authors should discuss the effect of the added magnetic particles and the applied magnetic field on the friction force.

Author Response

We sincerely thank the reviewers for their time and effort in evaluating our manuscript. We have carefully considered all comments and made necessary revisions to address these issues. Our responses are detailed below:

Reviewer Comment 1: For a, b and c in Figure 9, the authors just simply described the measurement results of the friction force in MRE footpads with and without a magnetic field. The authors should summary and analyze the measurement results clearly for readers' understanding.

Response 1: Thank you for your insightful comment. We have revised the manuscript to provide a clearer summary and analysis of the measurement results of the friction force in MRE footpads with and without a magnetic field. The revised section is as follows:

Overall, figures (a) to (f) in Fig. 9 show that the application of the magnetic field signifi-cantly reduces the friction in both the front and back legs, and this reduction in friction can be attributed to the arrangement of the magnetic particles within the MRE, which reduces the con-tact area and hence the friction. For example, in Fig. (a) and Fig. (b), during the anchor-pull phase, the friction of the front leg is 2.09 N without the magnetic field and 1.75 N for the back leg with the added magnetic field, which creates a friction difference between the front leg and the back leg, which makes the robot's back leg forward. Whereas, in the anchor push phase, the friction of the front leg is 1.90N and the back leg is 3.34N, which creates a friction difference which makes the front leg of the inchworm robot to extend forward. Similarly, Figs. 9 (c) and (d) comparisons demonstrate the friction measurements on the copper surface, where the friction of the front leg is 2.79N during the anchor pull phase, higher than that of the rear leg which is 2.64N, and during the anchor push phase, it is the rear leg which is 3.28N higher than that of the front leg which is 1.95 N. When on the wood surface, as shown in Figs. 9 (e) and (f), the friction of the front leg during the anchor pull phase is 2.68N, higher than that of the 2.63N for the rear leg, and in the anchor push phase, it was 3.89N for the rear leg over 1.96N for the front leg.The overall reduction in friction on the different surfaces proves the adaptability and effectiveness of the MRE foot-beds in a variety of environments.”

This text can be found on page 12, paragraph 2, lines 324-339.

Reviewer Comment 2: Authors should add a schematic diagram to explain the mechanism for the increasing of friction force under the magnetic field in part of 2.2.2. MREs Foot Integration.

Response 2: Thanks for pointing this out, we agree with this comment. We have added a schematic diagram in section 2.2.2 to explain the mechanism for the change in friction force under a magnetic field. The schematic illustrates how the magnetic field influences the alignment of magnetic particles within the MRE, thereby reducing the contact area and friction force. The detailed explanation accompanying the schematic is as follows:

Figures 6(b) and (c) show the mechanism of friction change in MRE footpads under mag-netic field, illustrating how the application of magnetic field affects the alignment of magnetic particles inside the MRE footpads, thereby reducing friction. By applying a magnetic field, the magnetic particles are aligned inside the MRE footbed in the direction of the magnetic field and the magnetic field magnetises these particles to create a magnetic force Fp between them and at-tract them to each other, leading to a reduction in the contact area and hence a reduction in fric-tion.

This addition can be found on page 8, paragraph 2, lines 248-259.

Reviewer Comment 3: Authors should discuss the effect of the added magnetic particles and the applied magnetic field on the friction force.

Response 3: Thanks for pointing this out, we agree with this comment. We have added a new section 3.3, titled "Discussion on MRE Properties and Their Impact on the Inchworm Robot Performance," to discuss the effect of the added magnetic particles and the applied magnetic field on the friction force. The discussion includes the following points:

The change in friction of the inchworm robot can be attributed to the response of magnetic particles in the magnetorheological elastomer to the applied magnetic field. Silicone materials usually have excellent wear resistance. The magnetic fillers added to silicone make the magne-torheological elastomer exhibit magnetorheological effect in the presence of a magnetic field, as well as controllable modulus and friction. It is worth mentioning that this characteristic is reversi-ble, that is, the magnetorheological elastomer can restore its original characteristics after the ap-plication of the magnetic field is stopped. Under the action of the magnetic field, the ferromag-netic particles in the elastomer matrix are arranged along the magnetic field lines to form an or-ganized chain structure, which reduces the maximum peak value of the microscopic surface of the MRE. The increased contact area reduces the friction of the magnetorheological elastomer foot pad under three different surface conditions in Figures 9(a) to 9(f).

This change in friction and magnetorheological elastomers help robots enhance environ-mental adaptability and motion efficiency. First, the magnetic particle arrangement induced by the magnetic field increases the contact interface, reduces random surface interactions and im-proves motion stability. Secondly, the dynamic response of MRE to the magnetic field can ad-just the friction characteristics in real time, allowing the robot to adapt to different surface condi-tions during movement. Finally, the reversibility of the magnetic field effect ensures consistent performance over multiple motion cycles. And the storage modulus and hardness of the isotropic magnetorheological elastomer will increase due to the addition of the magnetic field, which will increase the mechanical properties of the magnetorheological elastomer and make it more wear-resistant, which enables it to achieve more efficient energy transfer in the anchoring and anchoring stages of the inchworm robot.

The increase in the crawling speed of the inchworm robot is due to the combined effect of the mechanical properties of the magnetorheological elastomer and the controllable friction force. The crawling of the inchworm robot relies on the difference in friction between the front and back legs. By adding a reversible magnetic field to control the friction of the magnetorheological elastomer, the friction of the front and back feet of the inchworm robot can be changed. For example, under the condition of the copper plate, when no magnetic field is added, the friction of the hind legs is constantly greater than the friction of the front legs, with a difference of 0.49, and only anchoring can be performed, not anchoring. After adding the magnetic field, a greater fric-tion difference can be achieved, and the friction of the front legs and the friction of the hind legs can be controlled. The other two surfaces also show such performance. This helps the inchworm robot to move stably, and as shown in Figure 11, the crawling speed of the inchworm robot is sig-nificantly increased, because the increased friction difference enhances the motion performance of the inchworm robot.

This addition can be found on page 13, paragraph 2, lines 370-415.

Reviewer 2 Report

Comments and Suggestions for Authors

Dear authors!

Thank you for the scientific work presented. For the most part, there are no harsh comments. The text is written in understandable language. The results are quite acceptable.

1) What is the possible amount of backlash (free) movement of the moving components and are there ways to optimize the corresponding negative feedback system?

2) What are the additional physical and mechanical reasons (mechanisms) that influence the improvement of the environmental adaptability and the efficiency of the movement of the inchworm robot?

3) Figure 5a (a) Anchor pull and (c) Anchor push - are the same images, please reconsider 

Author Response

Thank you for your constructive comments and positive feedback about our work. We greatly appreciate your careful review and suggestions for improvement. Below are our point-by-point responses:

Comments 1: What is the possible amount of backlash (free) movement of the moving components and are there ways to optimize the corresponding negative feedback system?

Response 1: Thank you for raising this important question. We agree that component movement precision is crucial for robot performance. In our design, we have implemented several strategies to minimize backlash effects and optimize movement control. Through our mechanical design and control system, we have achieved stable locomotion performance. The specific measures include:

  1. Precise gait control using high-precision stepping motors.
  2. implementation of a simplified crank-slider structure to minimize free movement.
  3. Coordination of electromagnets and stepper motors through a central control unit to achieve precise matching of friction changes and crawling gait.
  4. The magnetorheological elastomer itself has the property of vibration isolation and damping.

Our experimental results show that these designs effectively support stable crawling motion. Although the analysis of the recoil motion of the moving parts may provide more insights for future optimization, our current measures have shown a clear effect in studying the friction changes during the crawling of the inchworm robot, allowing the friction test results to prove our point. Thank you very much for your question, which we will consider in subsequent inchworm robot research.

This text can be found on page 13 , paragraph 2, lines 370-415

Comments 2: What are the additional physical and mechanical reasons (mechanisms) that influence the improvement of the environmental adaptability and the efficiency of the movement of the inchworm robot?

Response 2: Thank you for this insightful question. We agree that it's important to comprehensively understand the factors affecting robot performance. Beyond the MRE characteristics, we have identified several key mechanical and control factors that contribute to environmental adaptability and movement efficiency. The revised section is as follows:

"In order to ensure that the inchworm robot can crawl stably on the surface, we selected a stepper motor that can accurately move forward to control the crawling gait of the inchworm robot. In addition, the crank slider structure of the inchworm robot is simple in design, which re-duces the free movement of the inchworm robot. The central control unit controls the electro-magnet and the stepper motor at the same time, so that the electromagnets of the front and hind legs of the inchworm robot can accurately change the friction of the magnetorheological elasto-mer during anchor pushing and anchor pulling. This stable and controllable friction change re-duces the energy loss caused by unnecessary surface interactions and reduces the possibility of slipping during crawling, thereby effectively improving the movement efficiency and environ-mental adaptability of the inchworm robot. Through experiments, it was found that this design enables the inchworm robot to adapt to the environment while achieving good motion performance."

These mechanical elements work in tandem with the MRE properties to improve overall performance. Because temperature also affects the friction of the MRE, we also controlled the temperature to 26 degrees Celsius during the experimental setup.

This text can be found on page 14 , paragraph 1, lines 405-415

Comments 3: Figure 5a (a) Anchor pull and (c) Anchor push - are the same images, please reconsider

Response 3: Thank you for pointing out this issue. We agree with this comment. Therefore, we have revised Figure 5 to show only two distinct phases: (i) anchor pull and (ii) anchor push, with their corresponding electromagnetic states and movement patterns clearly illustrated. The locomotion mechanism is now explained in detail with the following sequence:

"The locomotion capability is achieved through a coordinated sequence of actuations, as shown in Figure 5. In Figure 5(a), the crawling motion of the inchworm robot consists of two phases: anchor pull (i) and anchor push (ii). In the anchor pull phase (i), the electromagnets of the front legs are turned off, while the electromagnets of the rear legs are turned on. The friction of the magnetorheological elastomer pads of the rear legs is reduced, and the front legs maintain high friction for anchoring, thereby generating a friction difference between the front and rear legs. Then the crank-connecting rod mechanism starts to act, using the front legs as anchors to pull the rear legs forward.

In the anchor push phase (ii), the state of the electromagnets is reversed, the front leg electromagnet are turned on, and the rear leg electromagnets are turned off, making the friction of the front leg MRE pads less than the friction of the rear leg MR elastomer pads. The rear legs are anchors, and then the crank-connecting rod mechanism pushes the front legs forward. And return to the posture of the anchor pull phase. Then repeat the above actions, and this sequence of variable friction and mechanical drive allows the inchworm robot to crawl forward continuously.

As shown in Figure 5(b), the central control unit acts as the main controller, receiving power from the 9V power supply and controlling the mechanical drive and electromagnetic components of the inchworm robot. The mechanical drive is achieved through a stepper motor (powered by 24V), which accurately controls the movement of the front and hind legs through a crank-connecting rod mechanism. At the same time, the central control unit controls the electromagnet of the front and rear legs. The electromagnetic control is achieved through two parallel circuits, each channel containing a 24V power supply and a time relay system. The electro-magnets control the friction state of their respective MRE foot pads through magnetic field modulation, and the friction force of the magnetorheological elastomer pad is affected by the change of the electromagnet magnetic field."

This revision can be found on page 7, paragraph 2, lines 215-238]

Reviewer 3 Report

Comments and Suggestions for Authors

The authors have proposed an inchworm robot utilising friction change in MRE footpads under magnetic field influences.  The manuscript is well written; however, the following comments need to be addressed before it is considered for publication:

1.      Line 135, The title should be below the Figure 2.

2.    The mechanical and magnetic characteristics of the MRE need to be discussed.

3.      More SEM images with different magnifications are needed to understand the MRE structure.

4.      Need to explain Figure 5 in more detail.

5.      Sections 3.1 and 3.2 have the same heading (Friction measurement).

Author Response

We sincerely thank the reviewer for their constructive comments and suggestions that have helped improve our manuscript. We have carefully addressed all the comments and revised the manuscript accordingly. Below are our point-by-point responses:

Comment 1: The title should be below the Figure 2.

Response 1: Thank you for pointing this out. We have corrected the figure caption placement. The title is now placed below Figure 2.

This text can be found on page 4 , paragraph 2, lines 146-147.

Comment 2: The mechanical and magnetic characteristics of the MRE need to be discussed.

Response 2: Thank you for your insightful comment, we agree with this comment. We have added a comprehensive discussion of the mechanical and magnetic characteristics of the MRE in a new section 3.3 titled "Discussion of MRE characteristics and effects on the performance of the inchworm robot". This section detailed explanation is as follows:

"The change in friction of the inchworm robot can be attributed to the response of magnetic particles in the magnetorheological elastomer to the applied magnetic field. Silicone materials usually have excellent wear resistance. The magnetic fillers added to silicone make the magnetorheological elastomer exhibit magnetorheological effect in the presence of a magnetic field, as well as controllable modulus and friction. It is worth mentioning that this characteristic is reversible, that is, the magnetorheological elastomer can restore its original characteristics after the ap-plication of the magnetic field is stopped. Under the action of the magnetic field, the ferromagnetic particles in the elastomer matrix are arranged along the magnetic field lines to form an organized chain structure, which reduces the maximum peak value of the microscopic surface of the MRE. The increased contact area reduces the friction of the magnetorheological elastomer foot pad under three different surface conditions in Figures 9(a) to 9(f).

This change in friction and magnetorheological elastomers help robots enhance environ-mental adaptability and motion efficiency. First, the magnetic particle arrangement induced by the magnetic field increases the contact interface, reduces random surface interactions and im-proves motion stability. Secondly, the dynamic response of MRE to the magnetic field can ad-just the friction characteristics in real time, allowing the robot to adapt to different surface conditions during movement. Finally, the reversibility of the magnetic field effect ensures consistent performance over multiple motion cycles. And the storage modulus and hardness of the isotropic magnetorheological elastomer will increase due to the addition of the magnetic field, which will increase the mechanical properties of the magnetorheological elastomer and make it more wear-resistant, which enables it to achieve more efficient energy transfer in the anchoring and anchoring stages of the inchworm robot.

The increase in the crawling speed of the inchworm robot is due to the combined effect of the mechanical properties of the magnetorheological elastomer and the controllable friction force. The crawling of the inchworm robot relies on the difference in friction between the front and back legs. By adding a reversible magnetic field to control the friction of the magnetorheological elastomer, the friction of the front and back feet of the inchworm robot can be changed. For example, under the condition of the copper plate, when no magnetic field is added, the friction of the hind legs is constantly greater than the friction of the front legs, with a difference of 0.49, and only anchoring can be performed, not anchoring. After adding the magnetic field, a greater friction difference can be achieved, and the friction of the front legs and the friction of the hind legs can be controlled. The other two surfaces also show such performance. This helps the inchworm robot to move stably, and as shown in Figure 11, the crawling speed of the inchworm robot is significantly increased, because the increased friction difference enhances the motion performance of the inchworm robot.

In order to ensure that the inchworm robot can crawl stably on the surface, we selected a stepper motor that can accurately move forward to control the crawling gait of the inchworm robot. In addition, the crank slider structure of the inchworm robot is simple in design, which re-duces the free movement of the inchworm robot. The central control unit controls the electro-magnet and the stepper motor at the same time, so that the electromagnets of the front and hind legs of the inchworm robot can accurately change the friction of the magnetorheological elastomer during anchor pushing and anchor pulling. This stable and controllable friction change re-duces the energy loss caused by unnecessary surface interactions and reduces the possibility of slipping during crawling, thereby effectively improving the movement efficiency and environ-mental adaptability of the inchworm robot. Through experiments, it was found that this design enables the inchworm robot to adapt to the environment while achieving good motion performance."

This text can be found on page 13 , paragraph 2, lines 370-415

Comment 3: More SEM images with different magnifications are needed to understand the MRE structure.

Response 3: Thank you for this constructive suggestion. We agree that multiple magnification levels would provide a better understanding of the MRE structure. We have added microscopic observations at four different magnifications: 50×, 200×, 500×, and 1000×, which provide a comprehensive view from overall surface morphology to detailed particle distribution and interface bonding. The revised section is as follows:

"The surface microstructure of the magnetorheological elastomer was observed using a digital microscope (DVM6, Leica Microsystems, Germany) as shown in Figure 2. The micrographs were taken at magnifications of (i) 50×, (ii) 200×, (iii) 500×, and (iv) 1000×, and Figure 2(i) shows the homogeneous distribution of the surface of magnetorheological elastomers, and the same can be seen in Figure 2(ii). Magnetic particle agglomerates are found to be dispersed in the elastomer substrate and the particle density is clearly visible. Figure 2(iii) shows the spherical shape of the individual magnetic particles and their random embedding in the silicone substrate. Figure 2(iv) clearly demonstrates the interfacial bonding between the magnetic filler and the elastomeric base. Multi-scale microscopic observation demonstrates the structure of the magnetic filler and the base within the magnetorheological elastomer

The addition of a magnetic field affects the surface properties of magnetorheological elas-tomers, as shown in Figure 3, where micrographs were taken (a) without a magnetic field and (b) under the influence of a 60 mT magnetic field. A computer was connected to the digital micro-scope to store and process the image data."

Please see the attachment for pictures. This text can be found on page 4 , paragraph 2, lines 134-148.

Comment 4: Need to explain Figure 5 in more detail.

Response 4: Thank you for this important suggestion. We agree that a more detailed explanation of Figure 5 would improve clarity. We have expanded the description to comprehensively explain the locomotion mechanism and control system of the MRE-based inchworm robot. The added details are as follows:

"The locomotion capability is achieved through a coordinated sequence of actuations, as shown in Figure 5. In Figure 5(a), the crawling motion of the inchworm robot consists of two phases: anchor pull (i) and anchor push (ii). In the anchor pull phase (i), the electromagnets of the front legs are turned off, while the electromagnets of the rear legs are turned on. The friction of the magnetorheological elastomer pads of the rear legs is reduced, and the front legs maintain high friction for anchoring, thereby generating a friction difference between the front and rear legs. Then the crank-connecting rod mechanism starts to act, using the front legs as anchors to pull the rear legs forward.

In the anchor push phase (ii), the state of the electromagnets is reversed, the front leg electromagnet are turned on, and the rear leg electromagnets are turned off, making the friction of the front leg MRE pads less than the friction of the rear leg MR elastomer pads. The rear legs are anchors, and then the crank-connecting rod mechanism pushes the front legs forward. And return to the posture of the anchor pull phase. Then repeat the above actions, and this sequence of variable friction and mechanical drive allows the inchworm robot to crawl forward continuously.

As shown in Figure 5(b), the central control unit acts as the main controller, receiving power from the 9V power supply and controlling the mechanical drive and electromagnetic components of the inchworm robot. The mechanical drive is achieved through a stepper motor (powered by 24V), which accurately controls the movement of the front and hind legs through a crank-connecting rod mechanism. At the same time, the central control unit controls the electromagnet of the front and rear legs. The electromagnetic control is achieved through two parallel circuits, each channel containing a 24V power supply and a time relay system. The electro-magnets control the friction state of their respective MRE foot pads through magnetic field modulation, and the friction force of the magnetorheological elastomer pad is affected by the change of the electromagnet magnetic field."

This text can be found on page 7 , paragraph 2, lines 215-238.

Comment 5: Sections 3.1 and 3.2 have the same heading (Friction measurement).

Response 5: Thank you for identifying this inconsistency. We agree that different headings are needed to clearly distinguish between these sections. We have revised the heading of Section 3.2 to "Motion Performance Evaluation" to better reflect its distinct content and purpose.

This text can be found on page 12 , lines 345.
